# Nutrient Incorporation in First Feeding Seahorses Evidenced by Stable Carbon Isotopes

**DOI:** 10.3390/ani11020470

**Published:** 2021-02-10

**Authors:** Sonia Valladares, Miquel Planas

**Affiliations:** Department of Ecology and Marine Resources, Instituto de Investigaciones Marinas (CSIC), Eduardo Cabello 6, 36208 Vigo, Spain; mplanas@iim.csic.es

**Keywords:** seahorses, stable isotopes, food assimilation

## Abstract

**Simple Summary:**

Stable isotopes were used to assess the assimilation of food in early juvenile seahorses of *Hippocampus guttulatus* reared under two feeding conditions: *Artemia* or copepods. The results suggest that copepods are more efficiently assimilated than *Artemia* since higher growth and survival were related to copepods feeding. Also, the consumption and assimilation of preys by juvenile seahorses could be traced using stable carbon isotopes as the stable carbon isotope values in seahorses approached the values of the corresponding diet. To our knowledge, it is the first study to assess nutrient assimilation in a seahorse species using stable isotopes.

**Abstract:**

Nutritional issues are among the most critical factors in the initial survival of juvenile seahorses. Currently, there is a knowledge gap on the relationship between nutrient assimilation and the effects on initial mortalities and growth. In the present study, the stable isotope approach was used to assess the incorporation of two live preys (*Artemia* and copepods) in juvenile seahorses *Hippocampus guttulatus*. The changes in stable carbon isotope (δ^13^C) values were studied through two feeding experiments: feeding on *Artemia* or copepods (experiment 1), and shifting feeding from copepods to *Artemia* (experiment 2). In experiment 1, after 24–48 h of feeding, juvenile seahorses exhibited small but progressive changes in δ^13^C values towards those of the corresponding diet, indicating that the assimilation of the food offered was progressively enhanced from days 2–3. Similarly, in experiment 2, a diet shifting from copepods to *Artemia* caused an increase in δ^13^C values, reflecting a switch towards the isotopically enriched new diet (*Artemia* metanauplii). Differences in the assimilation efficiency of preys offered are discussed based on growth and survival rates. The enhanced growth performances and survivals achieved when the juveniles were fed on copepods could be related to higher efficient assimilation of copepods compared to *Artemia*. The present study demonstrates that the consumption and further assimilation of preys by juvenile seahorses could be traced using stable carbon isotopes. The research on nutrient assimilation of juvenile seahorses should enhance our knowledge on nutrient processes in developing seahorses for a better understanding of initial ontogeny in the early life stages of the species.

## 1. Introduction

The increasing demand for seahorses in traditional Chinese medicine, the aquarium trade and for curiosity is mostly supplied by wild-caught seahorses, leading to overexploitation and negative impact on natural populations [1,2]. In this context, seahorse culture represents an alternative to reduce fishing pressure on wild stocks [3,4]. Global interest in captive breeding of seahorses has increased in recent years to satisfy aquarium trade demand [2,5,6]. However, seahorse rearing is a relatively recent activity that faces many problems requiring convenient addressing [1,2,5,7]. In particular, low survival in early life stages is one of the main bottlenecks where research should be focused on.

Recent studies have dedicated much effort to study factors affecting the survival and growth of juvenile seahorses, mainly on prey type preferences, prey enrichment, temperature, light regime and rearing density [8,9,10,11,12]. Feeding is among the most decisive factors in determining initial survival in seahorse juveniles, especially regarding nutritional requirements, feeding efficiency, and digestive capabilities [13,14,15,16].

In seahorse rearing, the juveniles are generally fed on rotifers, *Artemia* nauplii, or copepods, depending on the species and the mouth size at the male’s pouch release [14]. *Artemia* is easily produced in large quantities but needs to be enriched to improve its nutritional quality. For some seahorse species, feeding of juveniles only on *Artemia* nauplii has resulted in poor survivals, probably due to their poor digestibility [13,16]. In such cases, an initial supply of copepods for a few days is a valid alternative to *Artemia*, enhancing survival and growth rates [10,14,16]. Despite recent improvements in the cultivation procedures and progress in survival rates, nutrient assimilation in first feeding seahorses is poorly known mainly due to difficulties in the quantification of feed intake, absorption and utilisation of nutrients. In this regard, studies should be undertaken to understand nutrient processes and to establish the most adequate diet for successful rearing in the early life stages of each seahorse species.

Faeces collection, direct gut content examination and physiological approaches are indirect techniques generally applied in studies related to the nutritional assessment in fish larvae [17,18]. However, those methodologies are difficult to apply in small specimens, like seahorses, at early life stages. Besides, these methods have some limitations since digested soft prey tissues are neither identifiable nor quantifiable. In such cases, direct tracing of dietary components becomes essential to achieve more precise information on feeding and effective nutrient assimilation. Radioactive isotopes have been used as reliable tracers to measure ingestion and assimilation rates [19,20]. However, their use is often impractical in nature because of health threats and potential environmental contamination. Alternatively, enriched stable isotopes have been recently applied to trace nutrients in fish larvae [21,22,23].

Stable isotopes represent natural tracers that allow the direct determination of ingestion and assimilation rates in organisms. The application of stable isotopes as a nutritional tool in aquaculture represents a powerful method to assess the incorporation of nutrients from the diet into consumers [24,25,26] since the isotopic composition of animal tissues is related to that of the consumer’s diet, with a slight variation named trophic discrimination [27,28]. The long-snouted seahorse *Hippocampus guttulatus* Cuvier 1829 is a European species whose rearing has been recently considered [12,15,29,30,31]. As in other seahorse species, significant improvements in husbandry and rearing techniques have been recently achieved [5]. However, the knowledge on essential issues is still limited, especially on those related to feeding and nutrition and their effects on initial mortalities in juveniles.

Due to the high importance of survival optimization in early juveniles and constraints related to both feeding and nutrition, and limitations/disadvantages of other methodological approaches available, we used the stable isotope approach to assess for the first time the feeding and assimilation of live preys in early juveniles of *H. guttulatus* when (1) initially feeding on *Artemia* or copepods and (2) shifting feeding from copepods to *Artemia*. The research on nutrient assimilation of seahorse culture is advisable to understand their feeding requirements.

## 2. Materials and Methods

### 2.1. Broodstock

Juveniles of the seahorse *H. guttulatus* Cuvier, 1829 were obtained from the broodstock maintained in ad hoc aquaria [30] at Instituto de Investigaciones Marinas (IIM-CSIC), in Vigo (Spain). Adult seahorses were kept under an annual temperature cycle ranging from 15 °C in winter to 19 °C in summer (±0.5 °C) and a natural photoperiod cycle (10L:14D in winter and 16L:8D in summer) [32]. Pumped seawater was filtered (5 µm) and UV treated, with a 10–15% water exchange rate per day. Water quality was checked periodically for NO_2_, NO_3_, and NH_4_/NH_3_ content (0 mg L^−1^) by using Sera Test Kits. Salinity and pH levels were maintained constant at 38 ± 1 and 8.1 ± 0.1, respectively. Captive adult seahorses were fed twice per day ad libitum on enriched adult *Artemia* (EG, Inve, Spain), and supplemented with captured mysidaceans (*Leptomysis* sp. and *Siriella* sp.).

### 2.2. Experimental Design

A batch consisting of 624 newborn seahorses, 0 DAR—days after male’s pouch release, was released by a male kept in captivity for 8 months. Immediately after male’s pouch release, the juveniles were randomly distributed (143–144 juveniles aquarium^−1^) into four 30 L aquaria. The rearing system was submitted to a constant 16L:8D photoperiod regime supplied by 20 W fluorescent lamps (Power Glo), a temperature of 20 ± 1 °C, a continuous inflow flux of 700 mL min^−1^ and gentle aeration in the upper part of the water column [15].

#### 2.2.1. Experiment 1: *Artemia* vs. Copepods Feeding

Juveniles were fed ad libitum until day 20 on two different feeding regimes (2 aquaria per treatment) consisting of:

*Artemia*: *Artemia* nauplii from days 0 to 10 (1 *Artemia* mL^−1^) and a mixture of *Artemia* nauplii and metanauplii (1:1; 1 *Artemia* mL^−1^) from days 11 to 20.

Copepods: a mixture of the calanoid *Acartia tonsa* and the harpacticoid *Tisbe* sp. (0.6 copepods mL^−1^) from days 0 to 20.

#### 2.2.2. Experiment 2: Shifting of Feeding Regime

Survivors in the copepods treatment of experiment 1 at 20 DAR were transferred to four aquaria (14–16 juveniles aquarium^−1^). Two aquaria received *Artemia* nauplii and metanauplii (1:1) until day 30, whereas the other two aquaria were maintained on copepods. The juveniles were fed *ad libitum* by increasing prey density according to juvenile growth.

*Artemia* nauplii were obtained by hatching EG cysts (Iberfrost, Tomiño, Spain) for 24 h at 28 °C. *Artemia* metanauplii were 24 h-enriched on a mixture containing the microalgae *Phaeodactylum tricornutum* (1.6·10^7^ cells mL^−1^), Spirulina (KF Iber Frost, Spain), and Red Pepper (Bernaqua, Belgium). Copepods were cultivated on a mixture of the microalgae *Isochrysis galbana* (10^7^ cells mL^−1^) and *Rhodomonas lens* (16^7^ cells mL^−1^).

### 2.3. Sampling and Data Analysis

At the onset of experiment 1 and before first feeding, 50 individuals were sampled to determine initial carbon isotope values, weight, and length. Also, samples (about 5 mg dry weight) of *Artemia* (nauplii and metanauplii) and copepods were collected at different experimental days, rinsed with distilled water, and kept frozen at −80 °C for isotope determination. Seahorse juveniles were also randomly collected before feeding time from each aquarium at different ages (1, 2, 3, 5, 8, 11, 15, 20, 25 and 30 DAR) for stable carbon isotope analysis (δ^13^C), weight and length. Sampled seahorses were anaesthetised with tricaine methane-sulfonate MS222 (0.1 g L^−1^), transferred individually to Petri dishes, photographed for standard length (SL) measurements, and weighed on a Sartorius microbalance (±0.01 mg). Then, seahorses were rinsed with distilled water and frozen at −80 °C. Standard length (SL) of juveniles was measured according to Lourie et al. [4] (SL = head + trunk + curved tail) from digital photographs using image processing software (NIS, Nikon). Daily mortalities were recorded throughout the feeding experiment. Live food and bulk seahorses were lyophilised before analyses.

One sample of juveniles (*n* ≥ 4 juveniles per sample) was taken per aquarium (two aquaria per treatment) at each sampling day. Except for isotopic analyses, all anaesthetised juveniles used for length and weight measurements (*n* = 10) were recovered after and transferred to their original aquaria (no mortalities were detected). Isotopic analyses in collected samples were run on pooled juveniles (*n* ≥ 4). Survivals were daily monitored on each aquarium and final survivals for each treatment were calculated using the final survivals for each duplicate aquarium. Each data point for each treatment corresponds to the mean ± standard deviation of two aquaria.

The isotope analysis was performed on sub-samples of 1 mg (bulk pooled juveniles) weighed into tin capsules. Carbon isotope analysis and elemental composition were performed at Servizos de Apoio á Investigación (SAI) of the University of A Coruña. Samples were measured by continuous-flow isotope ratio mass spectrometry using a FlashEA1112 elemental analyser (ThermoFinnigan, Italy) coupled to a Deltaplus mass spectrometer (FinniganMat, Bremen, Germany) through a Conflo II interface. Carbon stable isotope abundance was expressed as δ^13^C parts per thousand (‰) relative to VPDB (Vienna Pee Dee Belemnite) and Atmospheric Air, according to the following equation:δ^13^C (‰) = [(R_sample_/R_reference_) − 1](1)
where R is the corresponding ratio ^13^C/^12^C. As part of an analytical batch run, a set of international reference materials for δ^13^C (NBS 22, IAEA-CH-6, USGS-24) were analysed. The precision (standard deviation) for the analysis of δ^13^C of the laboratory standard (acetanilide) was ±0.15‰ (1-sigma, *n* = 10). Standards were run every 10 biological samples. The amount of carbon relative to the amount of nitrogen present in seahorses (C:N ratio) was also determined at different times as an indicator of nutritional variations due to endogenous and exogenous nutrient utilization. The isotopic analysis procedure fulfils the requirements of the ISO 9001 standard. The laboratory is submitted to annual intercalibration exercises (e.g., Forensic isotope ratio mass spectrometry scheme—FIRMS, LGC Standards, UK).

Due to the high content in lipids both in prey and seahorse juveniles, the original δ^13^C and C/N data were normalised for lipids using conversion equations specifically constructed for prey and syngnathids [33]. For that, the following equations were applied to δ^13^C values:Prey: δ^13^C_N_ = δ^13^C_F_/1.016Seahorse juveniles: δ^13^C_N_ = (δ^13^C_F_ + 1.166)/1.024, where δ^13^C_F_ corresponded to the isotopic values in frozen samples and δ^13^C_N_ was the normalised (lipid free) isotopic value.

Similarly, the following equations were used for C/N normalisation:Prey: CN_N_ = CN_F_/1.201Seahorse juveniles: CN_N_ = CN_F_/1.226,

Variables were checked for normality using the Shapiro–Wilk test. Growth parameters and survival between feeding regimes were compared through the Student’s *t*-tests. SPSS v.15.0 software (IBM, Armonk, NY, USA) was used to perform the statistical analyses at a significant level of *p* < 0.05.

### 2.4. Bioethics

Animal maintenance and manipulation practices were conducted in compliance with all bioethics standards of the Spanish Government (Real Decreto 1201/2005, 10th October 2005) and approved by the Bioethics Committee of IIM-CSIC. Sampled juveniles were anaesthetised or euthanised using tricaine methane-sulfonate (MS-222, Sigma, Aldrich, Germany) at a concentration of 0.1 mg L^−1^ or above.

## 3. Results

### 3.1. Growth and Survival

Newborn seahorses weighted 0.44 ± 0.03 mg dry weight and measured 15.30 ± 0.62 mm SL (Table 1). In experiment 1, final dry weight in 20 DAR juveniles fed on copepods (6.20 ± 1.07 mg) was significantly higher than in those fed on *Artemia* (4.42 ± 1.36 mg) (*t*-test = 2.97, *p* = 0.01) (Table 1, Figure 1A). Similarly, final standard length was also significantly higher when feeding on copepods than on *Artemia* (30.87 ± 1.89 and 27.96 ± 2.35 mm, respectively) (*t*-test = 2.84, *p* = 0.01) (Table 1, Figure 1C).

In experiment 2, the dietary shift from copepods to *Artemia* resulted in a lower, but not significant (*t*-test = 1.74, *p* = 0.11), mean dry weight in 30 DAR juveniles (7.46 ± 3.27 mg) compared to those from copepods regime (10.80 ± 3.34 mg) (Table 1, Figure 1B). The drop in weight for those juveniles were very likely due to selective mortality on the previous days (see Figure 2B). This feature is not uncommon in fish rearing at early stages. Alternatively, it could be an “artefact” related to sampling. Mean standard lengths of seahorses at 30 DAR did not differ significantly between treatments (35.29 ± 3.44 and 35.08 ± 2.19 mm for copepods and *Artemia*, respectively) (*t*-test = −0.85, *p* = 0.41) (Table 1, Figure 1D). Differences in growth between feeding regimes in experiments 1 and 2 were noticed considering weight gain rather than size gain (Table 1).

Regarding survivals, they were significantly different between feeding regimes (*t*-test = 48.91, *p* < 0.001) at the end of experiment 1 (20 DAR). Seahorses fed on *Artemia* exhibited very low final survival (7.87 ± 1.12%) in experiment 1 compared to those from copepods feeding (98.33 ± 2.36%) (Table 1, Figure 2A). Mortalities in juveniles fed on *Artemia* started from day 3 onwards (Figure 2A). After the dietary shift from copepods to *Artemia* (experiment 2), the average final survival from 20 to 30 DAR decreased slightly from 98.33 to 91.19%, whereas no mortalities occurred in juveniles continuously fed on copepods (Figure 2B). However, differences in final survivals between feeding regimes did not differ significantly (*t*-test = 3.62, *p* = 0.07).

### 3.2. Stable Isotopes

Considering stable isotopes in prey, δ^13^C values in copepods and *Artemia* nauplii were almost identical (−19.4‰) but lower than in *Artemia* metanauplii (−17.4‰) (Figure 3).

In experiment 1, the initial δ^13^C value in newborn seahorses was −17.3‰ (Figure 3A) and the values increased in both feeding regimes in the following 24–48 h. Thereafter, δ^13^C values in juveniles showed different trends depending on the feeding regime applied. In juveniles fed on copepods, δ^13^C values were rather stable until 20 DAR, ranging from −17.6 to −16.9‰ (Figure 3A). In *Artemia* regime, a stable trend was found until 11 DAR (−17.6 ± 0.4‰), but δ^13^C values increased progressively until day 20.

In experiment 2, δ^13^C values in seahorses fed continuously on copepods remained stable from 20 to 30 DAR (−17.7 ± 0.0‰) (Figure 3B), whereas a progressive increase in δ^13^C values occurred until 30 DAR (−14.6 ± 0.2‰) in seahorses fed on *Artemia* metanauplii (Figure 3B).

C:N ratios in *Artemia* nauplii and metanauplii were rather similar (4.3 and 3.9, respectively), but higher than in copepods (3.2). In experiment 1, C:N ratios dropped initially (0–2 DAR) in both treatments (Figure 4A) but differences between treatments were evidenced between days 5 and 8 (Figure 4A). From 11 DAR to 20 DAR, C:N ratios remained rather constant in seahorses fed on copepods (3.0–3.1) but decreased sharply (Figure 4A) in those fed on *Artemia* (from 3.3 to 2.8). In experiment 2, the values of C:N ratios from 20 to 30 DAR dropped similarly in both feeding regimes (from 3.1 to 2.9) (Figure 4B).

## 4. Discussion

Our study demonstrates the effects of dietary regimes on the early ontogeny of seahorses *Hippocampus guttulatus*. Juveniles fed on copepods showed significantly higher performance than those fed on *Artemia* (nauplii and nauplii + metanauplii). At the end of experiment 1 (0–20 DAR), the growth of seahorse juveniles fed on *Artemia* was significantly lower than in those from copepods regime. Similarly, in experiment 2 (20–30 DAR) lower growth was achieved when copepods were substituted by *Artemia* metanauplii. Furthermore, juveniles fed on *Artemia* suffered high mortalities in experiment 1. After a dietary shift from copepods to *Artemia* metanauplii at day 20 (experiment 2), survivals decreased slightly, whereas no mortalities occurred in juveniles maintained on the copepods feeding regime. The higher survivals and growth observed in copepods regime agree with previous studies on several seahorse species, including *H. guttulatus* [16,34], *H. subelongatus* [13], *H. trimaculatus* [10] and *H. reidi* [14,34]. Hence, the supply of copepods for at least a few days enhances both growth and survival in early developing juveniles of seahorses.

Growth and survival in fish larvae are related to efficient assimilation of food. The results from our study suggest that copepods are more efficiently assimilated by *H. guttulatus* juveniles than *Artemia*, at least during the first DAR. Initially (24–48 h), juveniles from both dietary treatments performed similarly, with δ^13^C values showing a trend similar to previous findings [31] but not related to δ^13^C values in prey. Afterwards, the trends diverged, indicating an improvement of food assimilation in both treatments. Early differences between dietary treatments in seahorse development would rely on two main facts. Firstly, copepods seem to better fulfil the nutritional needs of juveniles [14,34]. Secondly, the digestion capabilities on copepods are much higher than on *Artemia* nauplii [16], providing the required nutrients for tissue growth and resulting in higher survivals [34]. These findings agree with other studies that pointed out that *Artemia* passes largely indigested through the gut in early fish larvae [14,35]. Improved assimilation of copepods over *Artemia* nauplii, as a result of a higher digestibility in the former, was visually evidenced when analysing seahorse faeces, in which copepods generally appear fully digested, whereas *Artemia* nauplii may appear completely or partially intact. Chitinase activity was present in juvenile seahorses just after male’s pouch release but only increased significantly from 15 DAR onwards [16], when the specialization of the digestive tract initiates with the development of the first intestinal loop and mucosal folding [36].

Carbon stable isotope values (δ^13^C) are used to evaluate food assimilation as the isotopic composition of the carbon consumed by an individual represents the isotopic composition of the carbon assimilated into tissues [27]. Schlechtriem et al. [24] reported that δ^13^C values in common carp (*Cyprinus carpio*) larvae fed on two types of nematodes were influenced by the stable carbon isotope values of the food, indicating that the larvae assimilated nutrients from the nematodes. Food assimilation in *Piaractus mesopotamicus*, commonly known as pacu, larvae was also verified through the changes in the isotopic composition of the larvae [37]. In the present study, the change of δ^13^C values toward the values of the corresponding diet exhibited by juvenile seahorses in experiment 1 would indicate an increased efficiency of prey assimilation with age (depending on the prey type) from day 2–3 onwards. Those changes would also indicate when a diet shifting occurred. A pivotal difference between copepods and *Artemia* is the composition and permeability of their exoskeleton [38,39]. *Artemia* cuticle is thicker and difficult to break down by mechanical trituration, impeding its efficient digestion and assimilation. Besides, Randazzo et al. [34] reported differences in the digestion of the two types of live prey supported by the absence of supranuclear vesicles in the intestine of juveniles fed exclusively on *Artemia*. The results achieved by those authors through Focal Plane Array—Fourier Transform-Infrared (FPA-FTIR) Spectroscopy analysis evidenced changes in the biochemical composition (total lipids, phospholipids, and carbohydrates) of the liver, depending on the type of diet offered to seahorse juveniles.

In experiment 1, the fast increase observed in δ^13^C values from days 10 to 20 in juveniles fed on *Artemia* was related to more efficient assimilation of *Artemia* metanauplii (δ^13^C enriched with respect to copepods and *Artemia* nauplii) added from day 11. In experiment 2, the significant change in δ^13^C values after the diet shift from copepods to *Artemia* metanauplii would also reflect adequate assimilation of the new diet during that period (20–30 DAR) as the result of improved efficiency in digestion at those ages [36].

The above considerations on the different patterns of nutrient assimilation to synthesise organic tissue for growth of juvenile seahorses is supported by the values of C:N ratios, which reflect the number of fat reserves in tissues. The relative abundance of different stable isotopes in tissues is a consequence of their different involvement in chemical reactions. For example, the lighter isotope of carbon (^12^C) tends to form weaker bonds and react faster than the heavier isotope of carbon (^13^C). A decrease in total carbon reflects the consumption of endogenous lipid reserves. In this context, isotopic enrichment occurs because lipids are usually less enriched in ^13^C than other tissues, whereas an increase in carbon content might indicate the assimilation of exogenous nutrients along with the depletion in ^13^C [27]. Juvenile seahorses are active swimmers that start feeding immediately after birth due to the extremely reduced yolk reserves [36]. The decline in C:N ratios observed for the first 2–3 days in both feeding regimes (copepods and *Artemia*), accompanied by an increase in δ^13^C values, would confirm the use of endogenous lipid reserves rather than exogenous feeding on the administered prey, as reported by Blanco and Planas [40]. Hence, although seahorse juveniles show active foraging since the onset of first feeding, C:N ratios and δ^13^C values indicate that the efficient initial digestion of prey and nutrient assimilation be delayed for several days, depending on the prey considered (at least 2–3 or 5 days for copepods or *Artemia* nauplii, respectively). This statement agrees with Olivotto et al. [5] and Planas et al. [12], who pointed out a low feeding efficiency in first feeding *H. guttulatus* juveniles. The huge increase in C/N values until 10 DAR in *Artemia* nauplii feeding juveniles from experiment 1 was due to the high lipid content of nauplii. The improvement in the assimilation efficiency of *Artemia* with age after a shifting of the diet from copepods to *Artemia* (experiment 2) suggested by the increase in δ^13^C values, was also confirmed by the decrease in C:N ratios. Similar changes in C:N ratios and δ^13^C values were reported by Gamboa-Delgado et al. [25] in Senegalese sole (*Solea senegalensis*) larvae and postlarvae.

Efficient foraging is not necessarily accompanied by efficient digestion and assimilation of prey in young fish larvae [17]. This implies that the feeding schedule needs to be established especially for each species. In the case of *H. guttulatus*, the presence of poorly digested *Artemia* described in faeces of early developing *H. guttulatus* juveniles supports its low digestibility [16]. Limited growth and survival rates in seahorse juveniles fed on *Artemia* from first feeding were previously reported in *H. guttulatus* [15,34] and other species [8,34]. Although digestion and assimilation of *Artemia* result enhanced from a certain age, the initial feeding on copepods (co-feeding or not on *Artemia* nauplii) for at least several days will provide better results in terms of both growth and survival [34,41]. Otherwise, the initial low digestibility of *Artemia* and its limited nutritional quality would limit its contribution to tissue growth in juveniles, promoting higher mortalities, as demonstrated in experiment 1. Conversely, copepods are preferred over *Artemia* nauplii in first feeding *H. guttulatus* [40], providing essential nutrients with higher nutritional quality, and promoting higher survivals and growth rates [15]. Finally, considering that the diet of seahorse juveniles in their natural environment is primarily based on copepods [42,43,44], it is reasonable to consider copepods as the most adequate initial prey in rearing systems.

An interesting topic in trophic ecology or animal diet reconstruction studies is the application of trophic enrichment factors (TEFs) or isotopic discrimination factors, which correspond to the difference between the stable isotope ratios of a consumer and its food source [26,45]. When unknown, generalist TEF values can be used but it is recommended that they are experimentally derived whenever possible. For that, the isotopic equilibrium of consumers with diet must be reached. Discrimination factors and the time required to reach the equilibrium depends on diet type and quality, and other factors [46]). Our results revealed that the isotopic equilibrium for δ^13^C was only reached in newborns continuously fed on copepods (see steady-state in Figure 3B). Considering the isotopic values in 30 DAR seahorses (−17.7‰) and copepods (−19.4‰), the resulting discrimination factor for δ^13^C (∆^13^C) in juveniles continuously fed on copepods would be around 1.7‰.

## 5. Conclusions

The present study demonstrates for the first time that the assimilation of preys by *H. guttulatus* juveniles can be traced using stable carbon isotopes. In particular, C:N ratios and δ^13^C values were found to be very useful in detecting problems related to specific prey organisms in the initial feeding of seahorses. Considering the whole results achieved, the addition of copepods in the diet during the first feeding days is highly recommended for the early rearing (first five days) of *H. guttulatus* juveniles.

## Figures and Tables

**Figure 1 animals-11-00470-f001:**
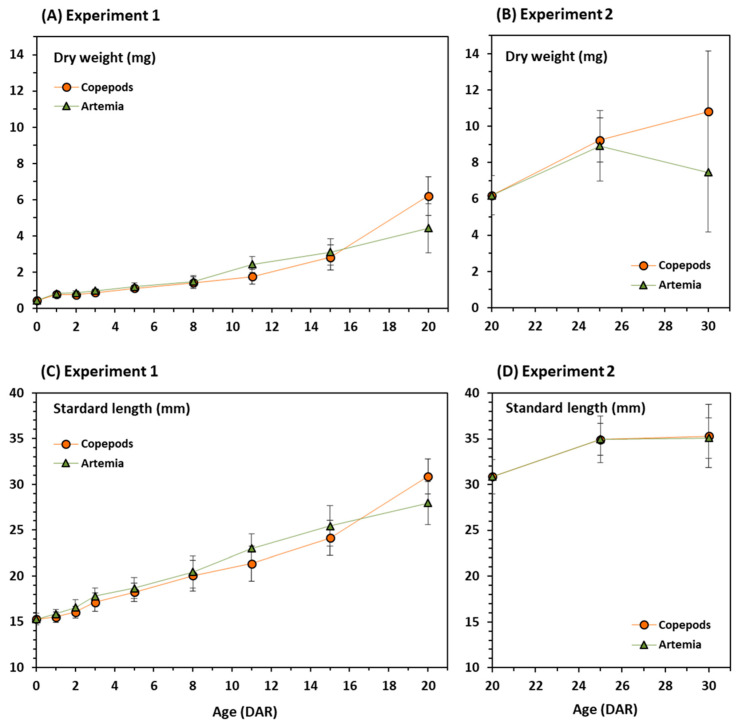
Growth ((**A**,**B**)—dry weight and (**C**,**D**)—standard length) of juvenile seahorses *Hippocampus guttulatus*. (**A**,**C**): Experiment 1—Juveniles fed on *Artemia* (nauplii and metanauplii) or copepods until 20 DAR; (**B**,**D**): Experiment 2—Juveniles fed on copepods in Experiment 1 until 20 DAR and subsequently fed on *Artemia* (nauplii and metanauplii) or copepods until 30 DAR. Means and standard deviations are represented.

**Figure 2 animals-11-00470-f002:**
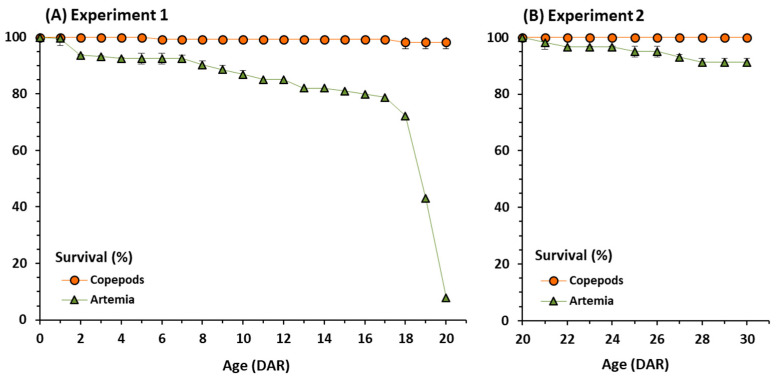
Survival (%) in juvenile seahorses *Hippocampus guttulatus* fed on copepods or *Artemia* over each experimental period. (**A**)—Experiment 1: Juveniles fed on *Artemia* (nauplii and metanauplii) or copepods until 20 DAR; (**B**)—Experiment 2: Juveniles fed on copepods in Experiment 1 until 20 DAR and subsequently fed on *Artemia* (nauplii and metanauplii) or copepods until 30 DAR. Means and standard deviations are represented.

**Figure 3 animals-11-00470-f003:**
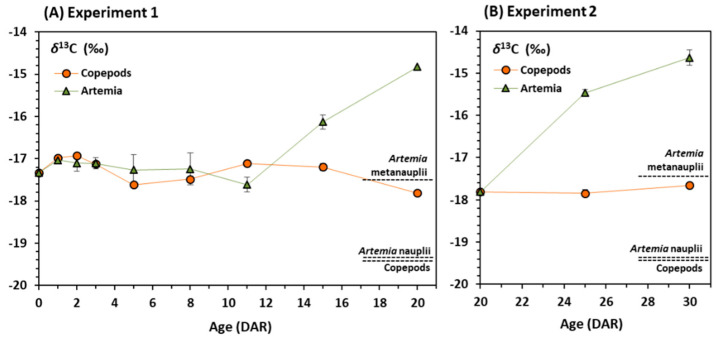
Changes in δ^13^C values in juvenile seahorses *Hippocampus guttulatus* fed on copepods or *Artemia* over each experimental period. (**A**)—Experiment 1: Juveniles fed on *Artemia* (nauplii and metanauplii) or copepods until 20 DAR; (**B**)—Experiment 2: Juveniles fed on copepods in Experiment 1 until 20 DAR and subsequently fed on *Artemia* (nauplii and metanauplii) or copepods until 30 DAR. Horizontal dotted lines represent the isotopic composition of prey (copepods and *Artemia* nauplii and metanauplii). Means and standard deviations are represented.

**Figure 4 animals-11-00470-f004:**
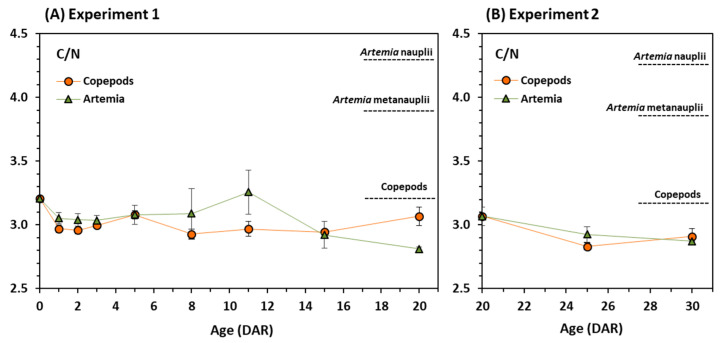
C:N ratios in juvenile seahorses *Hippocampus guttulatus* fed on copepods or *Artemia* over each experimental period. (**A**)—Experiment 1: Juveniles fed on *Artemia* (nauplii and metanauplii) or copepods until 20 DAR; (**B**)—Experiment 2: Juveniles fed on copepods in Experiment 1 until 20 DAR and subsequently fed on *Artemia* (nauplii and metanauplii) or copepods until 30 DAR. Horizontal dotted lines represent C/N values in prey (copepods and *Artemia* nauplii and metanauplii). Means and standard deviations are represented.

**Table 1 animals-11-00470-t001:** Survival, individual dry weight (DW), standard length (SL), and percentage of weight and size gain (WG and SG, respectively) of 0-, 20- and 30-days old seahorses *Hippocampus guttulatus* fed on two feeding regimes (means ± standard deviations).

Diet	Day	Survival (%)	DW (mg)	SL (mm)	WG (%)	SG (%)
Initial	0	100 ± 0	0.44 ± 0.03	15.30 ± 0.62		
Experiment 1: Days 0–20						
Copepods	20	98.33 ± 2.36	6.20 ± 1.07	30.87 ± 1.89	1409	202
*Artemia*	20	7.87 ± 1.12	4.42 ± 1.36	27.96 ± 2.35	1005	183
Experiment 2: Days 20–30						
Copepods	30	100 ± 0.00	10.80 ± 3.34	35.29 ± 3.44	2455	231
*Artemia*	30	91.19 ± 1.48	7.46 ± 3.27	35.08 ± 2.19	1695	229

## Data Availability

Data supporting the findings of this study are available from the author on request.

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
