# Peer review of "Nutrient Incorporation in First Feeding Seahorses Evidenced by Stable Carbon Isotopes"

_animals, 2021, doi:10.3390/ani11020470_

Round 1

Reviewer 1 Report

General comments:

I found it an interesting and original piece of work, contributing to knowledge on seahorse nutrition under culture conditions. It provides evidence of a sound methodological approach to the study of food assimilation, thereby enabling other research efforts in this field to follow.

However, my main concern relates to the interpretation of some of the results in the discussion. The authors state that changes in isotope values are evidence of an improved assimilation by sea horses as a consequence of including copepods in the diet. In reality, these results (Figures 3 and 4) only reflect that different foods were being assimilated equally well by sea horses as a direct result of the diet they were being fed. In other words, I think the authors are correct in concluding that “assimilation of preys by H. guttulatus juveniles can be traced using stable carbon isotopes”; but incorrect in assuring that their results on isotope analysis demonstrate that copepods are better assimilated than Artemia.

Sea horses fed the diet based on copepods indeed showed better growth and survival than those fed Artemia (although this was only true in experiment 1). Thus, results herein prove the assimilation of the copepod diet took place effectively, and that this appears to be a better diet considering seahorse growth and survival, two important zootechnical indicators. However, they do not provide proof that the copepod diet improved nutrient assimilation in juvenile seahorses compared to Artemia.

A second comment is on the absence of information regarding sample size for all analyses, both statistical and biochemical. It is not clear as to how they compared survival by means of a t-test (for which a n ≥ 2 is needed). Furthermore, it appears that the Shapiro- Wilk's tests used to check for normality may have such low power that it wouldn’t be possible to detect a non-normal distribution even if it were true. Data expressed as percentages (or proportions) are rarely normally distributed, hence my recommendation is to avoid hypothesis testing in this instance and whenever requirements of general lineal models are not met. It is also not clear the sample size used to compare growth amongst diets, or if samples of different dietary regimes were of similar size (balanced models). This is relevant when considering both normality and homogeneity of variance checks. Finally, it is necessary for readers to know how many subsamples were taken from each sea horse bulk prior to stable isotope analyses; whether this number was used to calculate the error bars for isotope values presented in Figure 3; and why error bars are not presented in Figure 4 if both C and N contents were obtained for each subsample.

A third comment is that authors only mention the existence of certain limitations associated to the use of stable isotope methodology in nutrition studies. Maybe a short explanation on these limitations would help to identify what exactly did the authors anticipate when they decided to use these methods, and how did such limitations affect their results and interpretations.

Specific comments:

Introduction:

Paragraph 4: it is not clear what is meant by “becomes essential to achieve more precise data research”. Consider rephrasing.

Experimental design:

Experiment 2: Where it says “The remaining juveniles in the copepods regime aquaria (20 individuals aquarium-1) were maintained on copepods feeding until day 30” it is necessary to clarify whether these juveniles were also transferred to two aquaria, resulting in four aquaria, two per treatment in experiment 2.

Results:

The number of replicates in all tests and statistic calculations needs to be stated if the readers are to interpret the results of these statistics adequately.

Stable isotopes:

Paragraph 2: This paragraph is a bit confusing. I suggest two different paragraphs: one describing δ13C values (Figure 3) and another describing C:N ratio values (Figure 4).

Paragraph 2: Is the following statement correct? “In the following 24-48 h, δ13C values in-creased in both feeding regimes but C:N ratios dropped in both treatments (Figure 4a).”? Figure 3 (both a and b) shows that δ13C values only increased in those dietary regimes based on metanauplii. There were ONLY two responses: higher values in sea horses fed Artemia metanauplii, and low values in those fed either Artemia nauplii or copepods.

Discussion:

Paragraph 4: “Afterwards, the trends diverged, indicating an improvement of food assimilation in both treatments. Early differences between treatments in seahorse development would rely on two main facts. Firstly, copepods seem to better fulfill the nutritional needs of juveniles [14,33]. Secondly, the digestion capabilities of copepods are much higher than for Artemia nauplii [16], providing the required nutrients for tissue growth and resulting in higher survivals [33].” These two facts could help to explain differences in growth and survival amongst seahorses fed with two different diets in experiment 1 (where these differences were indeed observed). But do not show an improved assimilation of one diet over the other. Results showed that sea horses feeding on copepods have an isotopic marker similar to that obtained in samples of the copepod diet; the same is true for those fed on Artemia metanauplii. I fail to see how these results evidence an improvement in food assimilation.

Paragraph 3: “In the present study, the change of δ13C values toward the values of the corresponding diet exhibited by juvenile seahorses in experiment 1 would indicate an increased efficiency of prey assimilation with age (depending on the prey type) from day 2 until day 20.” Alternatively, this change confirms that seahorses effectively assimilated the food they were given, be it copepods, Artemia nauplii or Artemia metanauplii, and when they were given it!

Paragraph 4: “In experiment 1, the increase observed in δ13C values from days 10 to 20 in juveniles fed on Artemia was related to a more efficient assimilation of Artemia metanauplii (δ13C enriched with respect to copepods and Artemia nauplii) added from day 11.” It might just be that seahorses consumed more metanauplii than nauplii, possibly because of differences in size of sea horses and prey (i.e. probability of encountering prey or some type of passive prey selection).

Last paragraph: “Finally, considering the fact that the diet of seahorse juveniles in their natural environment is primarily based on copepods [41,42], it is reasonable to consider copepods as the most adequate initial prey in rearing systems.” Many references (e.g. Manning, C.G., Foster, S.J., Vincent, A.C.J., 2019. A review of the diets and feeding behaviours of a family of biologically diverse marine fishes (Family Syngnathidae). Reviews in Fish Biology and Fisheries 29, 197–221. https://doi.org/10.1007/s11160-019-09549-z), enlist copepods together with mysid shrimps, decapod crustacean larvae and amphipods, amongst others, as part of the natural diet in juvenile seahorses. I suggest this information is included together with the appropriate references.

Tables and Figures

Legends of Tables and Figures: “Hippocampus guttulatus” should be in italics.

Reviewer 2 Report

It is a useful work providing more information on the critical stage of the sea-horse juvenile rearing. It is mostly nicely written. The results and figures need to be improved for clarity.

A drawback is that nothing is mentioned about lipid extraction and how lipids included in the food can affect the isotope values. Why you did not do lipid extraction in Artemia before you measure for stable isotopes? Artemia is expected to be high in lipids, especially compared to copepods. Or you could do lipid correction. Ideally, use both lipid extracted and not lipid extracted to see if there is a difference.

Why the isotope values do not reach an equilibrium, especially with Artemia? Normally the isotope values should be stable at some point since the juveniles are eating the same food. Is there probably something else influencing them?

You do not mention anything about the trophic discrimination/fractionation. Actually, you could calculate yourselves the trophic discrimination since you have a controlled experiment and you know the isotope values of the food and the consumer.

You could calculate the turnover rate, i.e. when (how many days) the seahorse obtains the new isotopic signature after being fed with a new food. You mention it somewhere in the discussion but you could emphasize that.

Abstract:

live prey not life prey

the description of the 2 experiments is not clear, in both cases a change is mentioned therefore it is not clear what is the difference between the experiments

Page 2, 2nd paragraph

since survival and growth rates result significantly enhanced.

Something seems wrong with the structure of this part of the sentence

Materials and Methods

Page 2

2.1. It is not necessary to mention again the full name of the species but it is enough to say H. guttulatus

Page 3, 1st paragraph

Salinity: I guess you mean ppt

2.2 consist of (not on)

DAR: the first time you give this term it is good to give the full words so that even people who don’t know it can understand it

Since you give information on the food of the metanauplii and copepods, why don’t you say something about the nauplii too?

Figure 1: The legend is a bit confusing.

Give A or I as subtitle to the figures

What are the bars standard error or deviation? Also, it’s not easy to distinguish the bars of each food

Species name in italics.

Say briefly what is each experiment or at least the second. The legend should be self-explanatory.

How do you explain the drop in the weight of the seahorses fed with Artemia? (Day 30)

Table 1:

Day 20 is included in both experiments. Why?

Figure 2: species name in italics

Results:

Was there any difference between the 4 aquaria or later 2?

Figure 3: it’s better to mention the word “Artemia” before Nauplii and Metanauplii as Copepods can also have nauplii. The figure legend should be self-explanatory

Why is there a sign of Artemia on the sign of Copepods on day 20;

Why there are no error bars in the experiment 2 but also in the specimens fed with copepods in the experiment one? Did all samples had exactly the same value?

Page 6, 1st line

How come the survival of 98% did not differ from the survival of 7.87%? Or do you mean something else? Anyway, it’s not very clear.

Page 7, last line from the end and in the Discussion (1st paragraph)

Artemia metanauplii: I thought they were fed in both nauplii and metanauplii? If it’s somewhere written wrong please correct accordingly

Discussion

Page 8, 2nd paragraph

Why δ13C of the juveniles of the first days do not resemble those of the food? Is it because they still have a similar value with the adult seahorse? Could you analyse some adult fish as well to compare?

Page8, 3rd paragraph

Stable isotope values

Page 9, 1st paragraph

FPA-FTIR: what is it? Please give the full names or a brief description

Page 9, 3rd paragraph

You explain the decrease in C/N for the first days but similar decrease occurred also later. What about this?

Page 9, 3rd paragraph

The sentence: “A decrease in carbon….isotope depletion [27]” I think needs improvement because the meaning is not directly straight-forward.

It is a bit confusing in the figures that the first (a) part refers to seahorses fed with copepods or Artemia and the second part (b) refers to the 2nd experiment that uses only the one type of seahorses of the experiment 1. You could denote this difference somehow or explain better in the legend.

Page 8, 2nd paragraph

How do you explain the difference between this and your older paper? (Valladares & Planas 2020) There you found the δ13C to decrease with growth and here the opposite although in both cases you started with Copepods and gradually gave more Artemia

General comment

The word Artemia is in some cases written in italics, in others not.

Round 2

Reviewer 2 Report

The manuscript has improved its clarity. Minor comments -based on the previous comments-follow.

Previous comment:

A drawback is that nothing is mentioned about lipid extraction and how lipids included in the food can affect the isotope values. Why you did not do lipid extraction in Artemia before you measure for stable isotopes? Artemia is expected to be high in lipids, especially compared to copepods. Or you could do lipid correction. Ideally, use both lipid extracted and not lipid extracted to see if there is a difference.

R:The original data have been corrected for lipids normalization using conversion equations (Planas et al., 2020; https://doi.org/10.3390/ani10122301), as indicated now in M&M (L. 185- 188). Figures and data on the text have been updated accordingly

New comment:

As far as I understand your other paper (Planas et al. 2020) refers only indirectly to lipid normalization conversion equations. The topic is different and no specific equations are used for normalization. Could you make it more explicit in the manuscript which equations you used (those for FR_ET?) and why?  

Previous comment: the description of the 2 experiments is not clear, in both cases a change is mentioned therefore it is not clear what is the difference between the experiments

R: That part has been rewritten. See new text in lines 124-128.

New comment

Do you mean you had 14-16 juveniles/aquarium in 2aquaria with copepods.

I guess you had 14 in one and 16 in the other that makes 30 specimens and you analysed them for isotopes twice that makes it impossible to have n ≥ 4 juveniles (line 160) per sample in each sample.

Either I didn’t understand something well or you are not precise in what you write.

Previous comment: How do you explain the drop in the weight of the seahorses fed with Artemia? (Day 30)

R: The drop in weight for those juveniles were very likely due to a selective mortality on the previous days (see figure 2B). This feature is not uncommon in fish rearing at early stages. Alternatively, it could be an “artifact” related with sampling.

New comment

Thank you for the explanation. You could add it in the manuscript too.

Previous comment: Table 1: Day 20 is included in both experiments. Why?

R: Day 20 does not correspond to experimental day but to the age of juveniles (DAR). As explained in M&M, experiment 1 was finished at 20 DAR, and the juveniles remaining at that age in the copepods treatment were immediately used (also 20 DAR) to carry out experiment 2..

New comment

Yes, but what food (Artemia vs. Copepods) did the juveniles eat that day?

Previous comment:

Was there any difference between the 4 aquaria or later 2?

R: Since the former description of the experimental design was not sufficiently clear to the reader, it is feasible that the reviewer did not understand the exact meaning of her question. There were 4 aquaria in experiment 2, which was set up using juveniles remaining in 2 aquaria from experiment 1 (aquaria from the copepods treatment). We hope that everything be clear on the new text (L- 124-128).

New comment: Sorry, my question was not formulated correctly. Was there any difference in the isotopic values and especially in growth parameters and mortality between each 2 aquaria with the same treatment? I am trying to see if there was anything else that influenced the experiment especially since the sample size in experiment 2 was low. Did you only have 1 sample per aquarium per analysis day? If I understand well, that makes 2 samples per analysis day which is rather low.

Previous comment:

Page 6, 1st line

How come the survival of 98% did not differ from the survival of 7.87%? Or do you mean something else? Anyway, it’s not very clear.

R: There should be a misunderstanding here. The paragraph clearly states that both treatment were significantly different) (“Regarding survivals, they were significantly different between feeding regimes (t-test = 48.91, p < 0.001) at the end of experiment 1 (20 DAR)”). L. 225-226.

New comment:

I refer to the lines: However, differences in final survivals between feeding regimes did not differ significantly (t-test = 3.62, p = 0.07). (Page 6, 1st paragraph, previous version, lines 238-239 new version).

Earlier you said: Regarding survivals, they were significantly different between feeding regimes (t-test = 48.91, p < 0.001) at the end of experiment 1 (20 DAR)”)

Please be clear where you refer each time. When you say ‘final’ do you mean at the end of experiment 2? Because experiment 1 also has different feeding regimes.
